# Remarks on Mitochondrial Myopathies

**DOI:** 10.3390/ijms24010124

**Published:** 2022-12-21

**Authors:** Patrizia Bottoni, Giulia Gionta, Roberto Scatena

**Affiliations:** 1Dipartimento di Scienze Biotecnologiche di Base, Cliniche Intensivologiche e Perioperatorie, Università Cattolica del Sacro Cuore, Largo Francesco Vito 1, 00168 Rome, Italy; 2Dipartimento Scienze Anatomiche Istologiche Medico Legali e dell’Apparato Locomotore—Sezione di Anatomia Umana, Università La Sapienza di Roma, Via Alfonso Borelli 50, 00161 Rome, Italy; 3Dipartimento di Medicina di Laboratorio, Madre Giuseppina Vannini Hospital, Via di Acqua Bullicante 4, 00177 Rome, Italy

**Keywords:** mitochondria, oxidative metabolism, electron respiratory chain, mutations, mitochondrial DNA, reactive oxygen species

## Abstract

Mitochondrial myopathies represent a heterogeneous group of diseases caused mainly by genetic mutations to proteins that are related to mitochondrial oxidative metabolism. Meanwhile, a similar etiopathogenetic mechanism (i.e., a deranged oxidative phosphorylation and a dramatic reduction of ATP synthesis) reveals that the evolution of these myopathies show significant differences. However, some physiological and pathophysiological aspects of mitochondria often reveal other potential molecular mechanisms that could have a significant pathogenetic role in the clinical evolution of these disorders, such as: i. a deranged ROS production both in term of signaling and in terms of damaging molecules; ii. the severe modifications of nicotinamide adenine dinucleotide (NAD)+/NADH, pyruvate/lactate, and α-ketoglutarate (α-KG)/2- hydroxyglutarate (2-HG) ratios. A better definition of the molecular mechanisms at the basis of their pathogenesis could improve not only the clinical approach in terms of diagnosis, prognosis, and therapy of these myopathies but also deepen the knowledge of mitochondrial medicine in general.

## 1. Introduction

In 1988, Wallace et al. showed that a mitochondrial DNA mutation was associated with Leber’s hereditary optic neuropathy. Specifically, this mutation converted a highly conserved arginine to a histidine at codon 340 in the NADH dehydrogenase subunit 4 gene [1].

The discovery of mutations in the mitochondrial DNA (mtDNA) led to an explosive expansion of research on mitochondrial myopathies. Over the past two decades, the rapid pace of identification of these clinically diverse disorders and their associated gene defects has left many physicians bewildered about the variety and complexity of these peculiar syndromes. Mitochondria are complex and fundamental organelles that play a central role in energetic metabolism of the cell. In facts, for many of the cells of our organism, mitochondria represent the powerhouse (i.e., the main source for ATP synthesis). This synthesis originates from an aerobic metabolism (i.e., fatty acids, carbohydrates, and amino acids are broken down to form CO_2_ and H_2_O). To realize this energetic process some fundamental steps must be implemented: (1) substrate transport; (2) substrate utilization by the Krebs cycle, beta-oxidation and so on; (3) electron transport chain; and (4) oxidative phosphorylation.

It is important to underline that this is only one of the interconnected functions of mitochondria which also regulate nucleotide and lipid synthesis, protein modification, calcium metabolism, free radical production and related signalling pathways, maintenance of the lipid membrane, fusion and fission activities, participation in immunity and programmed cell death.

Importantly, all organs/tissues which heavily rely on oxidative metabolism can show typical clinical manifestations related to energetic deficit (muscle, heart, nervous system, kidneys, endocrine organs).

On this basis, mitochondrial myopathies are a heterogeneous group of diseases mainly originating by the inability of mitochondria to sustain cellular energy demands due to structural and functional alterations of the whole oxidative metabolism apparatus.

Recently, the International Workshop of Experts in Mitochondrial Diseases defined mitochondrial myopathies as a group of progressive muscle conditions, primarily caused by the impairment of oxidative phosphorylation (OXPHOS). Myopathy is a typical manifestation of mitochondrial disorders because skeletal muscles show a high cellular energy demand. However, patients with mitochondrial myopathy often have dysfunction in different organs/tissues resulting in a high variability in clinical phenotype with significant influences on prognosis and, eventually, therapeutic approaches (Table 1) [2].

Last, but not least, clinical manifestations also depend on the number of mitochondria that harbour the alteration. In facts, mitochondria contain their own DNA, and this gives them peculiar genetic characteristics with significant clinical implications.

## 2. Epidemiology

Epidemiological studies on mitochondrial myopathies show the same typical difficulties because of the complexity of mitochondrial genetics with their intriguing genotype/phenotype inter-relationships. In fact, a single-point mutation in the mtDNA may produce isolated chronic progressive external ophthalmoplegia in one patient, and in another, a typical mitochondrial encephalomyopathy with lactic acidosis and stroke-like episodes. Yet, deletions in mtDNA may lead to either isolated ophthalmoplegia or to a more complex Kearns–Sayre syndrome. Moreover, some mitochondrial disorders, such as Leigh syndrome, may be caused by mutations in either mtDNA or a nuclear genome. In the case of Leigh syndrome, mutations have been identified in more than 30 different genes, and all show an effect on genes encoding proteins with fundamental roles in mitochondrial oxidative metabolism. Such a genetic and clinical heterogeneity causes problems in correct identification and diagnosis.

In spite of these difficulties, epidemiologic data seem to suggest that the mitochondrial diseases caused by all mtDNA mutations are not so rare. For example, studies realized in Northern England [3], Finland, Sweden, and Australia, showed disease prevalence (/100,000) of 6.57, 5.71, 4.7, 5.0, respectively [4,5]. Differently, 1 in 34,000 adults is affected by a mitochondrial disease due to nDNA mutation. Similar rates in children have been observed in Europe and Asia. On this basis, by combined biochemical, histologic, and genetic criteria, the prevalence of mitochondrial diseases in children is estimated to be 4.7–15 per 100,000. In the opinion of some epidemiologists, primary mitochondrial diseases affect 1 in 5000 people. Importantly, the real prevalence is probably underestimated because of the complexity of making a correct diagnosis in patients with multisystemic symptoms and with onset ranging from infancy to adulthood [4,5].

## 3. Genetics

Mitochondrial DNA (mtDNA) is a double-stranded circular molecule, composed of 16,569 base pairs. This DNA contains 37 genes: 13 encode polypeptides; 22 encode transfer ribonucleic acids (tRNA) molecules, and two encode ribosomal RNAs (rRNAs). The 13 polypeptide units are all components of the respiratory chain. Specifically, complex I contains seven subunits by mtDNA and 39 subunits derived from nDNA, complex III contains one mtDNA-derived subunit and 10 nDNA-derived subunits, complex IV consists of three mtDNA-derived subunits and 10 nDNA-derived subunits. Importantly, the four subunits of the complex II are all derived from nDNA.

This peculiar mtDNA shows some characteristics that differentiate it from nuclear DNA. These properties are responsible for the unusual genetic and clinical features of mitochondrial myopathies, specifically:aEach cell usually contains two copies of each autosome chromosome and a single copy of an X/Y chromosome (nDNA); however, each cell may contain hundreds to thousands copies of mtDNA, hence mitochondrial genoma is polyploid.bmtDNA is mainly maternally inherited.cmtDNA molecules are organized into discrete aggregates called nucleoids, which are probably linked to the internal mitochondrial membrane.dThis DNA lacks introns, so genetic information is more packed.eDue to its intrinsic characteristics and particular location, mtDNA undergoes spontaneous mutations more easily than nDNA

These peculiar aspects should be framed in the whole genetic physiology of mitochondria. In fact, nDNA encodes for almost 1700 mitochondrial genes, including over 200 respiratory chain proteins.

So, it has to be stressed that nDNA encodes the following: (1) most electron transport chain subunits and all ancillary proteins needed for proper subunit assembly; (2) factors needed for mitochondrial protein importation; (3) factors needed for mtDNA replication, transcription, and translation (“mtDNA maintenance”); (4) factors controlling the synthesis and assembly of phospholipids in the OMM and IMM; and (5) factors controlling mitochondrial dynamics, i.e., mitochondrial motility, fusion, fission, and mitophagy; (6) moreover, nDNA mutations, at the level of mitochondria, follow Mendelian genetics (inherited in an autosomal dominant, autosomal recessive, or X-linked pattern); (7) finally, de novo sporadic mutations in nDNA, at the mitochondria level, are also described [6,7].

This complex genetic milieu contributes to some significant functional and pathophysiological properties of mitochondria [8,9,10], such as:

*Mutation rate*. As already noted, mtDNA has a high mutation rate due to the lack of histones, the lack of introns and, above all, the potential presence of high concentrations of oxygen radical species in mitochondria.

*Mitochondrial heteroplasmy*. These organelles contain their own genomes. This causes a typical polyploidy which differently characterize each mitochondria/cell/tissue. Thereby, a mutation cannot affect all mtDNA, while it will hit just some mtDNA copies of a mitochondria/cell/tissue with significant implications for the phenotypic expression of a mutation.

*Maternal inheritance*. This is a typical feature of mtDNA, due to the physiology of oocyte fertilization, in which a sperm contains only a small quantity of mitochondria, with respect to oocyte (approximately 100 times less). In addition, the paternal mitochondria are both diluted by cell divisions and mitotic segregation and destroyed via apoptosis. Thus, mitochondria and the mitochondrial genotype of a fertilized egg are derived almost entirely from the mother. Moreover, the total mitochondria content in the primordial germ cell is randomly assigned into each primary oocyte. A rapid replication (i.e., amplification) occurs in each primary oocyte. In the end, each mature oocyte contains a different proportion of possible mutant mitochondrial DNA compared with the proportion found in the primordial germ cell. Importantly, random mutations of mtDNA can also occur in the germ cells, leading to offspring with different mtDNA genotype with respect to their mothers.

*Threshold effect.* As indicated above, considering mitochondrial heteroplasmy, not all cells in a tissue may have a mutation. Consequently, a minimal number of mutated mtDNAs must be present to allow the occurrence of the mitochondrial dysfunction. Hence, the clinical picture does not become evident until enough cells are affected, which in turn also depends on the importance of oxidative metabolism for the cell/tissue/organ. The threshold effect may vary between different tissues (brain, retina, muscles, heart, and kidney). Importantly, the mutation load in the cell/tissue/organ correlates with the severity of the disease and generally for overt disease, the mutation load is high (≥80 percent). Cells with high oxidative metabolism are heavily affected by mtDNA mutations; therefore, these disorders tend to affect disproportionately the brain and muscle (encephalomyopathies).

*Mitotic segregation*. This is the feature by which mitochondria are randomly distributed during cell division. This may cause variation in the amount of mutant mtDNA in a cell. The consequence is a possible modification of clinical phenotype when the level of mutant mtDNA overcomes the threshold for the specific tissue/organ.

*Postmitotic replication*. mtDNA replicates independently by cell cycle. This mtDNA replication in terminally differentiated cells (i.e., neurons or muscle cells) in response to specific stimuli (exercise, increased metabolic demand) may explain how the clinical symptoms can occur later in life.

Last but not least, it is also important to consider potential congenital (for example: genes: COQ2-7, PDSS1-2. and so on) or acquired derangements (for example, by drugs such as statins) of factors involved in CoQ synthesis, which may differently impair electron flux [2,4,7].

## 4. Hints on Main Mitochondrial Myopathies

### 4.1. Kearns–Sayre Syndrome (KSS)

The disorder is usually caused by single large-scale deletions of mtDNA. Respirometric studies in muscle showed severe combined defects of mitochondrial complexes containing mtDNA-encoded subunits. Particularly, the complex mainly damaged is Cytochrome c oxidase.

Interestingly, this syndrome is a multisystem mitochondrial disease characterized by a typical picture: onset before the age of 20, progressive external ophthalmoplegia, and pigmentary retinopathy. Moreover, at least one of the following alterations are also present: cardiac conduction block, cerebrospinal fluid protein greater than 100 mg/dL, cerebellar ataxia, short stature, deafness, dementia, and endocrine abnormalities. Importantly, KSS is a clinical subtype of chronic progressive external ophthalmoplegia [11,12].

### 4.2. Chronic Progressive External Ophthalmoplegia (CPEO) 

CPEO represents up to 20% of all mitochondrial disorders. Inheritance of CPEO can be sporadic, maternal, autosomal dominant, or autosomal recessive, because it can depend on both defects of mitochondrial and nuclear DNA. 

About 95% of patients have sporadic mtDNA point mutations or deletions/mutations in the *POLG* gene, which encodes the mitochondrial polymerase γ. Other described mutations include the following genes: *C10orf2*, *RRM2B*, *SLC25A4*, *POLG2*, *DGUOK* and *SPG7*, which cause multiple secondary mtDNA deletions.

These mutations cause abnormalities of respiratory chain function and/or dysfunction in mtDNA synthesis and/or in the maintenance of a balanced mitochondrial nucleotide pool. Patients show a slowly progressive paresis of extraocular muscles along with bilateral ptosis (mainly in the fourth decade of life). Diplopia is not constant and, often, is only transient. It is present as a slowly progressive weakness (paresis) of the muscles that control the eye movement (extraocular muscles). Finally, CPEO patients may present a pigmentary retinopathy, (“salt-and pepper” pigmentation) that, however, does not heavily affect vision [12,13]. 

### 4.3. Leigh Syndrome 

To date, incidence of Leigh disease should be approximately 1/40,000 births, though some studies show that the incidence is strictly population-specific (in a particular population in Quebec, Canada, it has been found an incidence of 1/2000 births, and in a Faroese population the incidence was 1/2500). 

Interestingly, the clinical picture of Leigh syndrome is caused by a variety of alterations of mitochondrial metabolism. Respiratory chain dysfunctions due to nuclear and/or mitochondrial DNA mutations, structural/functional modifications of the pyruvate dehydrogenase complex, may also be implicated. The syndrome is genetically heterogeneous, with pathogenic mutations identified in over 85 genes (mutations affecting subunits and assembly factors: complexes I, III, IV; tRNA mutations; coenzyme Q synthesis; mutations affecting synthesis and translation of mitochondrial DNA; mutations related to thiamine metabolism or fatty acids oxidation or glycogen metabolism).

The onset of Leigh syndrome is typically in infancy or early childhood causing brain abnormalities with ataxia, dystonia, external ophthalmoplegia (paralysis of the eye muscles), progressive neurodegeneration, developmental delays, seizures, lactic acidosis, vomiting, and hypotonia. Leigh syndrome, together with MELAS, represent the most common mitochondrial myopathies. The prognosis of Leigh syndrome is generally poor, with reduced survival after disease onset (a few months) [14,15].

### 4.4. Mitochondrial DNA Depletion Syndrome (MDS) 

MDS is a group of autosomal recessive disorders with a broad genetic and clinical spectrum, all characterized by a severe reduction in mtDNA content in affected tissues and organs. This reduction seriously hampers the synthesis of different subunits of mitochondrial respiratory chain complexes, therefore jeopardizing energy production. The onset of MDS is in infancy with a heterogeneous set of symptoms: muscle weakness, brain abnormalities, floppiness, eating difficulties, and developmental delays are common symptoms [16,17]. The prognosis is poor.

To date, it could be possible to distinguish four major types of MDS: 1Myopathic, caused by mutations in the TK2 gene (hypotonia and muscle weakness, facial weakness, bulbar dysarthria and dysphagia, elevated serum creatine phosphokinase);2Encephalomyopathic, caused by mutations in the SUCLA2, SUCLG1, or RRM2B genes, 2B, (hypotonia and pronounced neurological features);3Hepatocerebral, caused by mutations in the DGUOK, MPV17, POLG, or C10orf2 genes, (liver dysfunction and neurological disorders); and4Neurogastrointestinal, caused by TYMP mutations, (progressive disorders of gastrointestinal motility and peripheral neuropathy).

### 4.5. Mitochondrial Encephalomyopathy, Lactic Acidosis and Stroke like Episodes (MELAS) 

MELAS is a classic maternally inherited multisystemic disorder caused by mutations of mitochondrial DNA. Specifically, the m.3243A > G pathogenic variant in the mitochondrial gene *MT-TL* 1 is present in approximately 80% of cases, while 10 percent depend on an m.3271T > C tRNA mutation. Other reported mutations are in the genes MT-TQ, MT-TH, MT-TK, MT-TS1, MT-ND1, MT-ND5, MT-ND6, and MT-TS2. Intriguingly, cases of MELAS can originate as the result of a spontaneous mutation in mitochondrial genes and, thereby, are not inherited. 

MELAS typically occurs during childhood after a normal development. Relevant manifestations are: firstly, recurrent stroke-like episodes in the brain, leading to progressive and permanent neurological dysfunction and dementia, migraine-type headaches, vomiting, seizures, and lactic acidosis. Other symptoms may include PEO (progressive external ophthalmoplegia), muscle weakness, exercise intolerance, hearing loss, diabetes, and short stature [18,19].

### 4.6. Myoclonus Epilepsy with Ragged Red Fibers (MERRF) 

The prevalence of MERRF is likely to be less than 1 per 100,000 individuals. It is caused by mutations in mtDNA. About 80% of cases show an m.8344A > G mutation in the MT-TK gene encoding tRNA-Lys. Other mutations causing MERRF were discovered at level of the same MT-T gene (m.8356T > C, m.8363G > A, and m.8361G > A). 

Rare cases of MERRF syndrome may occur as the result of a spontaneous mutation. Thereby, these mutations are not inherited, but may be passed down to future generations if the affected individual is female. 

The reported mutations heavily derange the synthesis of the whole mitochondrial oxidative apparatus.

After a regular development, during childhood, a heterogeneous group of symptoms can occur, such as: myoclonus, epilepsy, ataxia, and muscle weakness. Additional symptoms are dementia, optic atrophy, bilateral deafness, peripheral neuropathy, spasticity, lipomatosis, and/or cardiomyopathy with Wolff–Parkinson–White syndrome [4,20].

MERRF is slowly progressive. Hence, the prognosis is poor.

### 4.7. Maternally Inherited Deafness and Diabetes (MIDD) 

The genetic basis is an A to G mutation at nucleotide position 3243 at the level of mtDNA encoding a tRNA. Interestingly, it causes the same mutation as that responsible for 80 percent of MELAS cases. Overlapping clinical pictures, in some patients, seems to suggest a continuum of expression for the *A3243G* mutation from diabetes and hearing loss alone to MELAS.

Notably, also this mutation causes a progressive decline in tissue OXPHOS.

MIDD typically occurs between the ages of 30 and 40. It is characterized by both a defect in insulin secretion, which progresses to insulin dependence, and sensorineural hearing loss. Other well-known anomalies are macular retinal dystrophy, myopathy, cardiomyopathy, renal disease, short stature, gastrointestinal disorders, and neuropsychiatric symptoms [21,22].

### 4.8. Mitochondrial Neurogastrointestinal Encephalomyopathy (MNGIE) 

Mutations in the *TYMP* gene (nDNA, 22q13.32-qter), encoding a protein involved in thymidine phosphorylation, causes MNGIE. These mutations, leading to total abolition of enzyme activity, cause thymidine and deoxyuridine accumulation in body fluids and tissues and, above all, imbalanced mtDNA replication and repair. Consequently, mtDNA show multiple deletions and/or partial depletion. MNGIE syndrome is inherited in an autosomal recessive mode. Orphanet reports just under 100 sporadic and familial cases [4,23].

MINGIE usually occurs between the first and fifth decade of life. This disorder causes ptosis, severe gastrointestinal motility disorders (visceral mitochondrial myopathy), dysphagia, gastroesophageal reflux, postprandial emesis cachexia, ophthalmoplegia and/or ophthalmoparesis, episodic abdominal pain, and diarrhea.

The prognosis of MNGIE is poor. In two studies of 35 and 102 patients, respectively, the mean ages at death were 35 and 38 years (range 15 to 58 years). 

### 4.9. Neuropathy, Ataxia, and Retinitis Pigmentosa (NARP) 

Mutations in the mitochondrial ATP synthase 6 gene causes NARP. Specifically, the large majority of patients has an m.8993T > C/G in subunit 6 of the mitochondrial H(+)-ATPase gene (MTATP6). Most NARP patients have 70–90% mutated mitochondrial DNA. Late-childhood or adult onset is usual. NARP is characterized by a great phenotypic variability and habitually manifests itself clearly in young adulthood. The typical clinical picture consists of learning difficulties, developmental delay, ataxia, ocular manifestations (ranging from sluggish pupils, nystagmus, ophthalmoplegia, night blindness, salt and pepper retinopathy to retinitis pigmentosa), and proximal neurogenic muscle weakness with sensory neuropathy. Other features may include short stature, seizures, corticospinal tract atrophy, depression, dementia, sleep apnea, hearing loss or cardiac arrhythmias. The prognosis is poor [4,24].

### 4.10. Pearson Syndrome (PS) 

PS is caused by single, large deletions of mtDNA, which can range from 1000 to 10,000 DNA building blocks (nucleotides). Generally, about 20–50% of patients show a “common deletion” with a length of 4977 bp. These mtDNA large deletions involve genes that codify proteins of oxidative phosphorylation. Although most PS cases are sporadic and caused by a somatic mutational event during early embryonic development, there are some reports of women with CPEO having children with PS. At the beginning, in the clinical picture, there are prevailing haematological symptoms, such as: severe anaemia, neutropenia, and thrombocytopenia. Then, gastrointestinal symptoms such as pancreas failure, vomiting, diarrhoea and feeding difficulties, muscular hypotonia, failure to thrive, vomiting and chronic diarrhoea. Intriguingly, episodic metabolic crises with lactic acidosis can occur. Pearson syndrome is usually fatal in infancy. Importantly, children surviving to the first years, often go on to develop Kearns–Sayre syndrome [25,26].

### 4.11. Iatrogenic Mitochondrial Myopathies

Mitochondria in general, and the proteins of the electron transport chain in particular, can be the innocent bystanders of some pharmacological treatments. The molecular mechanisms at the basis of these interactions are not well studied. A lot of drugs are capable of inducing myopathies and/or rhabdomyolysis with sometimes dramatic clinical implications (fibrates, thiazolidinediones, antipsychotics, amphetamines, antibiotics, FANS, and so on) [27,28].

Intriguingly, for fibrates, glitazones and metformin, some studies showed a neglected induced inhibitory effect on the activity of complex I of the electron respiratory chain. Importantly, this derangement not only hampers ATP synthesis, but it seriously impairs ROS homeostasis, NAD/NADH ratio, different signalling pathways related to ROS metabolism and, last but not least, causes a significant oxidative stress at cellular level (Table 2) [29,30,31,32]. 

## 5. Discussion

Interestingly, the main textbooks on neurology and fundamental articles about this topic, when dealing with the pathogenesis of mitochondrial myopathies, tend to limit the discussion to genetic aspects (mutations, deletion of a single protein), limiting the mechanism of disease to a generic reduction of ATP synthesis. Hence, this energetic failure deranges the physiology of the cell/tissue. 

However, this approach does not consider other fundamental functions of mitochondria in general, and of the electron respiratory chain in particular. As an example, mutations of proteins related to mitochondrial oxidative metabolism may induce not only a reduction of synthesis of ATP, but also significant modifications in ROS production and/or NADH/NAD ratio that directly and/or indirectly may have significant roles, not only in pathogenesis, but also in the evolution of these disorders. 

Our manuscript aims to push researchers to better consider the molecular pathophysiology of these disorders, as well as from a biochemical point of view, in which ATP shortage is just the most evident alteration of mitochondrial myopathies. 

In fact, in our opinion, an accurate definition of the molecular structural/functional derangements of mutated/deleted mitochondrial proteins may ameliorate the understanding of the pathophysiology of different mitochondrial myopathies with potential development in therapeutic approaches. Last but not least, this understanding could shed new light on different aspects of mitochondrial molecular medicine, a topic that is becoming increasingly important in the pathophysiology of different human diseases.

## 6. Conclusions

To date, mitochondrial myopathies represent a heterogeneous group of diseases characterized by a serious energetic deficit in ATP due to a derangement of mitochondrial oxidative metabolism. Often, they are an epiphenomenon of a more complex derangement of tissues and organs that heavily rely on oxygen for their energetic purposes. However, this energetic deprivation does not fully justify the whole complex and heterogeneous clinical picture, above all in terms of symptoms and evolution, of these diseases. In fact, a congenital or acquired mitochondrial derangement causes more important serious disruption of the whole cell physiology than a simple energetic deficit. From a metabolic point of view, an impaired mitochondrial oxidative metabolism induces severe modifications of nicotinamide adenine dinucleotide (NAD)+/NADH, pyruvate/lactate, and α-ketoglutarate (α-KG)/2- hydroxyglutarate (2-HG) ratios that, for example, may promote epigenetic modifications in the cell. Moreover, a deranged electron respiratory chain securely impacts the physiology of ROS metabolism with implications in terms of damage of cellular components and altered signaling pathways [30,31,32,47,48,49].

## Figures and Tables

**Table 1 ijms-24-00124-t001:** Typical mitochondrial myopathies that are generally multisystem disorders.

Myophathy	Pathogenesis	Inheritance	Age	Mitochondrial Target	Main Symptoms	Prognosis
Kearns-Sayre syndrome (KSS)	single large-scale deletions of mtDNA	generally, not inherited. (sporadic), rare case of mitochondrial, autosomal dominant, or autosomal recessive	before the age of 20	Mainly cyt C oxidase	progressive external ophthalmoplegia, and pigmentary retinopathy. cardiac conduction block, cerebrospinal fluid protein greater than 100 mg/dL, cerebellar ataxia, short stature, deafness, dementia, and endocrine abnormalities	slowly progressive disorder. Prognosis related to level of organs involvment. arrythmias
Chronic progressive external ophthalmoplegia (CPEO)	Deletion/mutation of mtDNA (i.e., tRNA at nucleotide 3243 in which there is an A to G), or nuclear genes: *POLG*, *C10orf2*, *RRM2B*, *SLC25A4*, *POLG2*, *DGUOK*, *SPG7*	sporadic, mitochondrial, autosomal dominant, or autosomal recessive	Aroud 40s years	defective function of oxidative phosphorylation	Ptosis, Limited eye movements, and Hearing loss, Mild muscle weakness, dysphagia, cataracts	prognosis depends on the associated features,
Leigh syndrome	Different pathogenic mutations identified in over 85 genes	nuclear or mtDNA mutations.	Generally, infancy and childhhod	Dysfunction of pyruvate dehydrogenase complex and oxidative phosphorylation	Mainly developmental delay or psychomotor regression failure to thrive, weakness/hypertonia, ataxia, oculomotor palsy, seizures, lactic acidosis	generally poor
Mitochondrial DNA depletion syndrome (MDS)	Different mutations in the TK2,SUCLA2, SUCLG1, RRM2B, DGUOK, MPV17, POLG, C10orf2; TYMP genes	Maternal and autosomal recessive	newborns, infants, children, or adult	different subunits of mitochondrial respiratory chain complexes	Different clinical pictures: Myopathic; encephalomyopathic; hepatocerebral; neurogastrointestinal	generally poor
Mitochondrial encephalomyopathy, lactic acidosis and stroke like episodes (MELAS)	mtDNA: m.3243A > G, gene *MT*-*TL* (80% of cases) and m.3271T > C tRNA mutation (10%)	maternally inherited	childhood	tRNA and NADH dehydrogenase	stroke-like episode, hemiparesis, hemianopia, or cortical blindness. focal or generalized seizures, recurrent migraine, vomiting, short stature, hearing loss, and muscle weakness.	poor
Myoclonus epilepsy with ragged red fibers (MERRF)	A-to-G transition at nucleotide 8344 (m.8344A > G) of the *MT*-*TK* genetRNA (Lys)	Spontaneous mutations, maternally inherited	Childhood, adolescence or early adulthood	oxidative phosphorylation	myoclonus, epilepsy, ataxia, myopathy, dementia, optic atrophy, deafness, peripheral neuropathy, spasticity, cardiomyopathy with WPW syndrome.	Generally poor. It can depend on age, severity of symptoms, organs involved.
Maternally inherited deafness and diabetes (MIDD)	mutation in mtDNA gene MT-TL1, encoding tRNA for leucine, and in rare cases in MT-TE and MT-TK genes, encoding tRNAs for glutamic acid, and lysine, respectively.	maternally inherited	mean age of onset is 30–40 years	defective function of oxidative phosphorylation	Diabetes, deafness, Chorioretinal abnormality, Dyschezia, Macular dystrophy, Malabsorption, Cerebellar hypoplasia, arrhythmias, heart failure, ophthalmoplegia, Muscular weakness,	prognosis for MIDD is better than that for MELAS
Mitochondrial neurogastrointestinal encephalomyopathy (MNGIE)	Mutations of TYMP gene (nDNA)	autosomal recessive	range from 5–60 years of age	defective function of oxidative phosphorylation	gastrointestinal disorders (dysphagia, cramping, vomiting, diarrhea, gastroparesis intestinal pseudo-obstruction) related to abnormal bowel motility. Neurological symptoms includes chronic progressive ophthalmoplegia, sensorimotor peripheral neuropathy	progressive degenerative disorder with a poor prognosis
Neuropathy, ataxia, and retinitis pigmentosa (NARP)	More frequent: m.8993T > C/G subunit 6 of mt ATPase gene	maternally inherited	Childhood	defective function of oxidative phosphorylation	sensory neuropathy, muscle weakness; ataxia, retinitis pigmentosa, developmental delay, seizures, dementia, deafness, arrhythmias.	poor prognosis

**Table 2 ijms-24-00124-t002:** Some iatrogenic mitochondrial myopathies.

	Mitochondrial Target	Mitochondrial Derangement	Clinical Disorders	References
Nucleoside analogues	Mitochondrial polymerase	mtDNA synthesi inhibition	Hepatic steatosis, lactic acidosis, myophaty, neuropathy, nephrotoxicity	[33,34]
Gentamicin, chloramphenicol, tetracycline	mtDNA	mtDNA synthesi inhibition	Deafness, renal failure, myopathy	[35,36]
Metformin	Complex I	Inhibition of NADH-ubiquinone oxidoreductase	Lactic acidosis, myophathy	[35,37]
Fibrates (clofibrate, gemfibrozil, fenofribate, etc)	Complex I	Inhibition of NADH-ubiquinone oxidoreductase I	Myopathy and rhabdomyolysis	[30,38]
Thiazolidinediones (pioglitazone, troglitazone)	Complex I	Inhibition of NADH-ubiquinone oxidoreductase	Liver failure, rhabdomyolysisr	[30,39,40]
doxorubicin	mtDNA	Mutations inducer	Cardiomyopathy	[41,42]
Cisplatin	mtDNA	Mutations inducer	Cardiomyopathy	[34,35,36]
Corticosteroids	Complex I	Inhibition of NADH-ubiquinone oxidoreductase	Myophathy	[34,43]
Local anaesthetic (bupivicane, lidocaine)	ATP synthase	Inhibition of complex V and oxidative phosphorylation	Myophathy	[34,35,36]
Propofol	Coenzyme Q	Inhibition of electron transport at CoQ level	rhabdomyolysis, heart failure, hepatomegaly, asystole	[36,44]
Statins (simvastatin, cerivastatin, etc)	Coenzyme Q; complex I	Inhibition of electron transport at level of complex I and CoQ	myopathy, rhabdomyolysis	[34,45]
Beta-blockers (metoprolol, propranol)	Complex I	Inhibition of NADH-ubiquinone oxidoreductase	myophathy	[46,47]

## Data Availability

Data from PubMed—National Library of Medicine.

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
