# Peer review of "Remarks on Mitochondrial Myopathies"

_ijms, 2022, doi:10.3390/ijms24010124_

Round 1
Reviewer 1 Report
This paper “An Update on Mitochondrial Myopathies” is well written, interesting to a readership working in the field of neuromuscular disorders. Methodologically correctly set, clear goals, and the references correctly follow the entire article. However, from my point of view, less informative is the table 1. Unfortunately, there are some limitations which preclude this paper from being published in the current status a mayor revision.
I have some recommendations:
1. What is the main update addressed by this research?
2. Which specific gap in the field of mitochondrial myopathies do you want to address with this update?
3. What does this update add to the area of mitochondrial myopathies compared with other published material?
4. A discussion section is lacking!

Author Response
Reply to review.
First of all, we want to thank reviewer for your positive assessment.
With reference to his remarks. We insert a discussion paragraph containing all the recommendations of the reviewer.
At last, considering “…less informative is the table 1..”. We want to clarify that with put this simple table to facilitate the reading and understanding of a text which deal with a topic classically considered “orphan diseases”. Moreover, a table could be useful to editorial purposes.
We remain to your disposal
Best regards
Roberto Scatena

Reviewer 2 Report
This manuscript is another revision of the different myopathies associated with mitochondrial dysfunction. Although authors include many diseases related with different mutations in mitochondrial DNA some other myopathies associated with nuclear genes codifying mitochondrial proteins are not included. If authors do not introduce these myopathies such as Q-synthesis dependent myopathies, ancillary protein defects, Iron motility defects such as splastic paraplegia and fusion/fission defects the title must be changed to "An update on mitochondrial-DNA dependent myophaties.
1.- If authors want to maintain the title, incorporation of nDNA-dependent myopathies must be included in this update.
2.- In introduction, the anabolic activities of mitochondia such as nucleotide synthesis and protein modification and lipid synthesis must be also included in the activities of mitochondria.
3.- Is Epidemiology section correct? The subject of this review is to update the knowledge about mitochondrial myopathies. The section only includes the complex scenario to reach a correct diagnostic of these diseases. A more focussed section about the incidence of these diseases is more correct.
4.- In the section about genetics, lines 113 and following, factors involved in CoQ synthesis must be also included in the list of nDNA encoding genes that produce mitochondrial-related diseases.
5.- In the section about the hints of main mitochondrial myopathies, authors must define PEO (in line 236) because only CPEO has been defined previously and also correct the line 252, it seems that is a line not completely finished.
6.- Section about iatrogenic mitochondrial myopathies must be revised in deep since it seems that the treatment with many known pharmacological compounds produce mitochondrial damage but the incidence of these adverse effects is not included in this section. As nDNA-dependent mitochondrial myopathies have not been included in this revision, this section does not fit with the aim of the manuscript.
7.- Figure 1 is very poor. Information provided by this image is near nothing. Only the components of the mETC and the location of ROS production and nothing more. This figure is not acceptable for this publication.
Author Response
- According to reviewer’s suggestion, we modified title. We want to stress that for editorial purpose (this article is not strictly directed to neurologists), we preferred to stress the most important mitochondrial myopathies, from a clinical point of view, as considered by Harrison’s Texbook of Medicine, Cecil’s Textbook of Medicine, Current diagnosis & treatment neurology, Bradley and Daroff’s Neurology in Clinical Practice, 8th Ed.
- Morerover , we include in introduction “…The other anabolic activities of mitochondria ..”
- Unfortunately, there are few epidemiological studies on this topic (anyway all reported in the second paragraph) probably because these disorders are considered “orphan diseases”.
- According to reviewer’s suggestion, we put into section, factors involved in CoQ synthesis
- PEO is being labeled
- “Section about iatrogenic mitochondrial myopathies”. We partially agree with reviewer about the fit of the section, but this is a fundamental and neglected aspect of mitochondria pharmacology and, albeit rarely, it caused dramatic cases of rhabdomyolisys (as cited in the text and reported in references)
- According to reviewer, we removed figure 1

Round 2
Reviewer 2 Report
Only a minor changes, the sections of pages 7 and 8 because is seem that section about MERRF is not finished (line 258) and section about MIDD is in other format.
Author Response
Dear Sirs,
we have modified manuscript according to reviewer's suggestions (two typing errors)
Regards
Roberto Scatena